# Multi-Target Alternative Approaches to Promoting Fresh-Cut Carrots’ Bioactive and Fresh-like Quality

**DOI:** 10.3390/foods11162422

**Published:** 2022-08-12

**Authors:** Carla Alegria, Elsa M. Gonçalves, Margarida Moldão-Martins, Marta Abreu

**Affiliations:** 1SFCOLAB—Associação Smart Farm COLAB Laboratório Colaborativo para a Inovação Digital na Agricultura, Rua Cândido dos Reis nº1, Espaço SFCOLAB, 2560-312 Torres Vedras, Portugal; 2cE3c—Centre for Ecology, Evolution and Environmental Changes & CHANGE—Global Change and Sustainability Institute, Faculdade de Ciências, Universidade de Lisboa, 1749-016 Lisbon, Portugal; 3Unidade Tecnologia e Inovação, Instituto Nacional de Investigação Agrária e Veterinária, I.P.. Av. da República, Quinta do Marquês, 2780-157 Oeiras, Portugal; 4GeoBioTec Research Institute, Campus da Caparica, Universidade Nova de Lisboa, 2829-516 Monte da Caparica, Portugal; 5LEAF—Linking Landscape, Environment, Agriculture and Food Research Center, Associated Laboratory TERRA, Instituto Superior de Agronomia, Universidade de Lisboa, Tapada da Ajuda, 1349-017 Lisbon, Portugal

**Keywords:** abiotic stress treatments, heat shock, UV-C, micro-perforated packaging film, microbiological development, bioactive compounds, sensorial quality

## Abstract

Fresh-cut fruits and vegetables, as near-fresh foods, are a quick and easy solution to a healthy and balanced diet. The rapid degradation of nutritional and sensory quality during the processing and storage of a product is critical and plant-type-dependent. The introduction of disruptive technological solutions in fresh-cut processing, which could maintain fresh-like quality with less environmental impact, is an emerging research concept. The application of abiotic stress treatments (heat shock and UV-C) induces metabolic responses and microbial effects in plant tissues, potentially slowing down several quality senescence pathways. The previously selected combined and single effects of heat shock (100 °C/45 s; in the whole root) and UV-C (2.5 kJ/m^2^) treatments and two packaging conditions (oriented polypropylene (OPP) vs. micro-perforated OPP films) on controlling critical degradation pathways of fresh-cut carrots and on promoting bioactive and sensory quality during storage (5 °C, 14 days) were studied. Among the tested combinations, synergistic effects on the quality retention of fresh-cut carrots were only attained for applying heat shock associated with micro-perforated OPP film packaging. Its effects on reducing (3.3 Log_10_ CFU/g) the initial contamination and controlling microbiological spoilage (counts below the threshold limit of 7.5 Log_10_ CFU/g), increasing the bioactive content (38% and 72% in total phenolic content and chlorogenic acid, respectively), and preserving fresh quality attributes prove to be a viable alternative technology for shredded carrot processing.

## 1. Introduction

Fruit and vegetable consumption is considered important for health, with recommendations from the scientific community and government dietary guidelines promoting an increase in their consumption for a healthy and balanced diet and lifestyle [1]. Fruits and vegetables provide essential vitamins, minerals, and other phytochemicals that can contribute to smooth functioning of the human body and can help protect from some non-communicable diseases [2].

Fresh-cut (FC) fruits and vegetables help meet consumer needs, and the FC market still continues to grow in developed countries, as they are convenient and ready-to-eat and retain close to fresh-like nutritional and sensory quality [3,4,5]. However, environmental sustainability (e.g., excessive water and chemical usage) and product safety issues (risk of pathogen contamination and growth) [6] in the FC industry have come into the spotlight. These drawbacks are primarily associated with minimal processing operations [5].

Maintaining the quality and safety of fresh-cut produce is complex, depending mainly on its associated microbiological load and knowledge of crop-specific critical quality deterioration pathways. It is well established that fresh produce can contain a high level of contamination after harvest. Fresh-cut processing promotes rapid deterioration due to tissue damage: stress caused by peeling and cutting the fruits and vegetables into different shapes significantly increases tissue respiration, leading to biochemical deterioration such as browning, texture breakdown, off-flavours, and a risk of microbial development [7]. Furthermore, contamination with both spoilage and pathogenic microorganisms may occur during fresh-cut processing, during which the released nutrients promote its growth [8], and to the best of our knowledge, in FC fruits and vegetables, no effective treatments to control microbial outgrowth exist [6,9,10].

Fresh-cut carrot products, such as shredded, sliced, stick, and baby carrots, are among the most frequently consumed ready-to-eat vegetables worldwide [11]. Particularly in the shredded format, carrots undergo rapid microbial degradation, quickly reaching unacceptable limits (aerobic colony counts 10^6^–10^8^ CFU/g [12]) and impacting their characteristic aroma [13]. On the other hand, the high loss in essential phytochemicals (e.g., phenolics and carotenoids) by leaching in washing solutions compromise the product’s nutritional and bioactive value, and the fresh-like colour of the product [14,15].

Developing new, effective, non-invasive, and non-chemical techniques for improving and maintaining quality is the industry’s timely question. Fresh-cut technology combines several treatments during the product’s shelf-life: sanitisation treatments during processing, modified atmosphere packaging, and refrigeration conditions during storage. In this strategy, combining low-intensity, conventional, or emergent treatments provides a series of preservation factors (barriers) that microorganisms cannot overcome while preventing drastic changes in fresh-like quality. Furthermore, combining treatments with different targets (multi-target preservation) as an alternative to just combining low-intensity single-target treatments may benefit this preservation strategy.

Heat shock and UV-C radiation, considered abiotic stresses by nature, can improve product quality, safety, bioactivity, sensorial attributes, and shelf-life [16]. Plant tissue responds to abiotic stresses (e.g., wounding, temperature, and radiation) by activating defence mechanisms, inducing *de novo* synthesis of phenolic compounds, and interfering in tissues’ physiology by changing respiratory activities and ethylene synthesis [17,18]. Thus, FC quality retention can be achieved by modulating these induced stress responses depending on the intensity of the abiotic treatments and the nature of the specific plant tissues [19,20]. 

Heat treatments (immersion in hot-water baths) effectively reduce microbial levels and, under suitable time/temperature conditions, prove to be a practical methodology for controlling microbial growth in several FC fruits and vegetables [21,22,23]. As a stress treatment, it also impacts several biochemical pathways, including heat-shock protein synthesis and disruption of normal cellular protein synthesis, namely quality-related enzymes (PPO, POD, PAL, and PME) [24,25,26,27,28]. 

Recent findings have shown that using UV-C radiation can induce the expression of antioxidant enzymes, increasing the biosynthesis of vitamins, phenolic compounds, and carotenoids [29,30,31,32,33] and improving the antioxidant capacity of FC products [34,35,36]. UV-C treatments have also been investigated as an alternative sanitising methodology to ensure fresh-cut product safety and to reduce the microbial loads on the surfaces of FC fruits and vegetables [34,36,37,38,39].

In FC carrots, using a heat shock (100 °C/45 s) prior to a shredding operation offers significant microbial reduction and better microbial control during shelf-life, decreased respiration rates, and fresh-like quality maintenance [15,16,28]. It was also recently proven that such heat shock promoted phenolic biosynthesis and accumulation to levels exceeding raw content [27]. Regarding the use of UV-C treatments, several studies in FC carrots have reported an increase in phenolic [20,35,40,41] and carotenoid contents during storage [16].

The present study evaluated the combined effects of eco-friendly treatments (heat shock and UV-C) and MAP conditions on controlling FC shredded carrots’ key degradation pathways and on promoting its bioactive and sensory quality. The hurdle concept is explored to the fullest within this research by integrating previously selected abiotic stress treatments (heat shock: 100 °C/45 s; UV-C: 2.5 kJ/m^2^), single or combined, with two MAP solutions (OPP vs. micro-perforated OPP films) on the quality of fresh-cut carrots during storage (5 °C, 14 days). 

## 2. Materials and Methods

### 2.1. Materials

Carrots (*Daucus carota* L. cv. Nantes, 80 kg) were obtained from a local fresh-cut processor (Campotec, Torres Vedras, Portugal) and transported to the lab in a refrigerated truck (5 °C ± 2 °C) in commercial stackable box containers (600 × 400 × 100 mm). Upon arrival to the laboratory, the carrots were hand-sorted (removal of damaged units), washed (50 mg/L NaOCl), dried, and maintained at 5 °C (± 1 °C; FitoClima S600, Aralab—Equipamentos de Laboratório e Electromecânica Geral Lda., Rio de Mouro, Portugal) until use (16 h).

For packaging, oriented polypropylene (OPP) films (Amcor Flexibles Neocel—Embalagens Lda., Lisbon, Portugal) were provided by the same fresh-cut processor (Campotec, Torres Vedras, Portugal).

Commercial standards of chlorogenic acid (CAS 327-97-9) and β-carotene (CAS 7235-40-7) were purchased from Sigma Chemical Co. (St. Louis, MO, USA). HPLC-grade solvents were purchased from Panreac (Barcelona, Spain), and analytical grade solvents were purchased from Panreac (Barcelona, Spain) and Sigma (St. Louis, MO, USA).

### 2.2. Sample Preparation

Of the carrot lot, 10% was used (Table 1) for raw material characterisation. The remaining carrots were equally divided and used to prepare treated (sample id: H, U, and HU) and untreated (sample id: C) samples, as shown in Figure 1.

#### Minimal Processing, Abiotic Stress Treatments, and Storage

Minimal processing was conducted in a sanitised room using sanitised apparatuses. The carrots were peeled using a sharp stainless steel vegetable peeler, treated according to their experimental design (H, U, HU, and C), and shredded (Dito MV-50 equipped with a CX.21, knife J7-8, Dito Sama, Aubusson, France). Treated (H, U, and HU) and control (C) samples were packaged in portions of 125 g using bags made from films of 35 μm bioriented polypropylene (200 × 110 mm) (Amcor Flexibles Neocel—Embalagens Lda., Lisbon, Portugal). Two films were used: film A (commonly used in FC carrot production) and film B (an alternative packaging solution). The permeability of film A had oxygen (OTR) and carbon dioxide (CTR) transmission rates of 1100 cm^3^/(m^2^·24 h·atm) and 3000 cm^3^/(m^2^·24 h·atm) at 23 °C, respectively. Film B (PPlus®—35PA120) had the same characteristics as film A, plus laser-established micro-perforations at 120 µm intervals. The bags were heat-sealed (Tish-400 impulse sealer, Tew Electric Heating Equipment Co, Ltd., Taipei, Taiwan) and stored at 5 °C (FitoClima S600, Aralab—Equipamentos de Laboratório e Electromecânica Geral Lda., Rio de Mouro, Portugal) for 14 days. For each tested abiotic stress treatment and packaging condition, ten bags per sampling day (0, 3, 5, 7, 10, 12, and 14 days) were prepared to comply with analytical procedures: 5 bags for physical-chemical, biochemical, and sensorial analysis and 5 bags for microbiological determinations.

Regarding the control (C) samples, no other treatment was applied to the carrots except the wounding stress (peeling and shredding) and packaging procedures (Figure 1).

The heat-shock (H) samples were treated following previously optimised conditions of 100 °C/45 s [15,16,25,28] and according to Figure 1: carrots were peeled and immersed in a thermostatically controlled pilot-scale steriliser set to 100 °C (±1 °C; monitorisation with Ellab T-type thermocouples). The treatment time (45 s) started immediately after carrot immersion. After the heat treatment, the carrots were placed in ice-cold water (5 min), dried, and further processed (shredding and packaging procedures).

The UV-treated samples were prepared as shown in Figure 1. The UV-C treatments were conducted in a temperature-controlled environment (5 °C) using a FitoClima S600 chamber (Fitoclima, Aralab—Equipamentos de Laboratório e Electromecânica Geral Lda., Rio de Mouro, Portugal) equipped with a UV-C reflector rack. The reflector rack was equipped with three TUV 18 W unfiltered germicidal emitting lamps (59 × 2.8 cm, λ = 254 nm, Phillips, Eindhoven, The Netherlands), placed at 14 cm intervals and a distance of 12.5 cm from the chamber walls. The lamp rack was placed at a distance of 20 cm above the illumination area, maximising product exposure to the UV-C source. Prior to use, the lamps were turned on and allowed to stabilise (15 min). UV-C dose (kJ/m^2^) was monitored with an HD2102.2 photoradiometer (Delta Ohm, Padua, Italy) equipped with an LP471 UVC probe (Delta Ohm, Padua, Italy) placed on the illumination area. The UV-C treatment was carried out with the chamber door closed to provide UV protection for the operator. The carrots were peeled and immediately shredded. The shredded carrots were distributed in a single layer on a tray covered with a reflector layer (food grade aluminium foil) to maximise UV light exposure and treated with the UV-C dose of 2.5 kJ/m^2^ [42]. After treatment, the UV-C-treated shredded carrot samples were packaged (film A and film B) and stored.

The samples for the combined treatment of heat shock and UV-C (HU) were prepared by applying the single treatments sequentially according to the previously described conditions and illustrated in Figure 1.

### 2.3. Analytical Procedures

#### 2.3.1. Headspace Gas Analysis

Headspace gas samples (*ca*. 2 mL) were taken with a hypodermic needle through an adhesive septum previously fixed on the sample bags and were analysed for oxygen and carbon dioxide concentrations (%) using a Checkmate 9900 O_2_/CO_2_ gas analyser (PBI-Dansensor, Ringsted, Denmark). The gas analyser uses a zirconium sensor for O_2_ determination and an infrared detector for CO_2_ detection. After approximately 30 s, the O_2_ and CO_2_ percentages from the package headspace are displayed. Five bags (sample replicates) were analysed per sample type on each sampling day.

#### 2.3.2. Total Phenolic Content

The total phenolic content (TPC) was determined by the Folin–Ciocalteu assay as described elsewhere [25]. The extraction was performed in five sample replicates (three independent measures per sample replicate), and the total phenolic content was expressed as mg chlorogenic acid equivalents per 100 g of fresh tissue (mg CAE/100 g).

#### 2.3.3. Chlorogenic Acid Quantification

The TPC assay extracts were used for the quantification of chlorogenic acid by HPLC-DAD according to [43] with the modifications described in [19]. Chlorogenic acid was quantified from an external standard curve (5–100 µg/mL).

#### 2.3.4. Antioxidant Capacity

Antioxidant capacity (AOx) was determined according to the procedure described by [44], using the ABTS radical cation decolourisation assay. The same extracts prepared for the TPC assay were also used for the AOx assay. A standard curve was prepared using Trolox as a reference, and the results were expressed as mg Trolox equivalents per 100 g of product (mg TEAC/100 g).

#### 2.3.5. Phenylalanine-Ammonia Lyase (PAL) Activity

The PAL activity assay was adapted from [45] with the modifications described in [19]. PAL activity was defined as the amount (μmol) of t-cinnamic acid synthesised per hour using a t-cinnamic acid standard curve (0–0.15 μmol/mL). Five sample replicates were analysed in triplicate.

#### 2.3.6. Total Carotenoid Content

The total carotenoid content (TCC) was determined according to [46]. In semi-dark conditions, 2 g of the fresh sample was mixed with 20 mL of an acetone/ethanol (1:1) solution containing 200 mg/L butylated hydroxytoluene (BHT). After homogenisation (20,000 rpm × 1 min, Yellow line DI 25 Basic polytron), the homogenate was filtered (Whatman #4 filter) and washed with the extraction solution until no further colour change was observable. The filtrate volume was adjusted (100 mL), and 50 mL of n-hexane was added. The mixture was shaken, and after phase separation, 25 mL of nanopure water was further added and allowed to stand until complete phase separation. An aliquot of carotenoid solution (hexane phase) was transferred into a glass cuvette, and spectrophotometric readings at 470 nm were collected (ATI Unicam UV/VIS 4 spectrophotometer). A standard curve was prepared using β-carotene as the standard (0–15 mg/L), and the results were expressed as β-carotene equivalents per 100 g of product (mg β-carotene eq/100 g). As for previous determinations, five replicates were analysed in triplicate.

#### 2.3.7. pH and Soluble Solids Content

The pH was measured (room temperature) in the homogenised carrot samples in distilled water (1:1, *w*:*v*) using a pH meter (Crison Micro pH 2001, Crison Instruments, Barcelona, Spain). The soluble solids content (SSC), expressed as °Brix, was measured from the previous homogenate using a digital refractometer (DR-A1, ATAGO Co Ltd., Tokyo, Japan) equipped with a thermostatic water bath set at 20 °C. One measurement per sample replicate was made for each determination.

#### 2.3.8. Sensorial Analysis

The panel members (8 trained panellists) were already familiarised with the product and scoring system from previous studies [15,16,28]. The samples were served on white dinner plates marked with three-digit codes and presented in a randomised order. The panellists were asked to evaluate the samples during storage, scoring the difference in perceived intensity between the sample and the fresh reference (freshly untreated shredded carrots) concerning overall acceptance to determine the product sensorial rejection index using a five-point hedonic rating scale: 1 = excellent freshly cut, 2 = good, 3 = limit of usability, 4 = poor, and 5 = unusable. The rejection index’s mean scores were calculated, and scores above 3 (limit of usability) indicated unacceptable samples.

#### 2.3.9. Microbiological Responses

The total mesophilic aerobic count (TAPC) was performed according to [47]. Yeasts and moulds (Y&M) were determined using a Rose Bengal Chloramphenicol Agar with surface inoculation and incubated at 25 °C for 5 days. The lactic acid bacteria count (LAB) was assessed according to [48]. The microbial counts were expressed as Log_10_ CFU/g.

#### 2.3.10. Statistical Analysis

Data from the trial were subjected to analysis of variance (ANOVA) using the Statistica™v.8 Software from Statsoft (Tulsa, OK, USA) [49]. Statistically significant differences (*p* < 0.05) between samples were determined according to the Tukey Honestly Significant Difference (HSD) test. For principal component analysis (PCA), all variables were mean-centred and standardised (scaled) to unit variance prior to analysis (correlation matrix). The principal components were obtained by computing the study data correlation matrix’s eigenvalues and eigenvectors [50]. A score for each sample was calculated as a linear combination for each quality variable for each PCA component. The contribution of each variable to the PCA score was deduced from the parameter loading for the factor. A bi-dimensional representation of this multi-dimensional set was made for the principal components that accumulated a significant percentage of original information, above 70%, which is considered sufficient to define a good model for qualitative purposes [51]. A hierarchical cluster analysis was run: the clustering process involved data standardisation, assessment of a dissimilarity measure among samples, and the use of a grouping technique. The Euclidean distance was used as a dissimilarity distance and Ward’s method as the grouping technique.

## 3. Results and Discussion

We used a principal component analysis (PCA) to explore the effects of the tested abiotic stress treatments and packaging films on the quality of fresh-cut shredded carrots during storage. A preliminary PCA was performed to ascertain which (quantitative) variables contributed to the model (Appendix A Appendix A). According to low factor loadings, the variables total carotenoid content (TCC), antioxidant capacity (AOx), soluble solids content (SSC), and yeast and mould counts (Y&M) were excluded from the model, and its discussion is held separately.

Hierarchical cluster and principal component analyses were used to establish a relationship between the abiotic stresses, MAP conditions, and FC carrot quality evolution during storage (Δ0–14 days). The resulting data matrix contained 56 samples and 9 variables (the codes are provided in Appendix A ). The data set included eight categories of samples identifying untreated (control; C), heat-treated (H), UV-treated (U), and combined-treated (HU) samples packaged in two film types (A and B). The quantitative variables corresponded to gas composition (O_2_ and CO_2_), microbiological data (TAPC and LAB), pH, total phenolic content (TPC), chlorogenic acid content (CA), PAL activity (PAL), and sensorial rejection scores (Rejection) (the mean values are shown in Appendix A Appendix A).

The PCA explained 80.4% of the original data variability in the first two dimensions (Figure 2). The PC1 accounted for the highest proportion of variation (50.8%) in the data and was most heavily loaded with microbiological counts, pH, sensorial rejection, and headspace gas composition (Table 2).

In Figure 2a, the variable vector projection on PC1 indicates that sensorial rejection was significantly dependent on microbial development (TAPC and LAB) negatively loaded with PC1. Lactic acid bacteria development contributed heavily to FC carrot quality deterioration, leading to product rejection, as illustrated by LAB’s and Rejection vector projections. It is also suggested that excessive LAB growth and product acidification (directly correlated) are promoted when the atmosphere inside the package exhibits high CO_2_ and low O_2_ (CO_2_ and O_2_ vector projection).

Several studies in different FC carrot formats (baby, slices, sticks, and shredded) demonstrated that gas composition significantly influences the product’s sensory quality, impacting product decay, particularly microbial growth [7,28,52,53,54,55]. Barry-Ryan et al. [7] stated that lower O_2_ levels rather than higher CO_2_ levels are the main triggers of this spoilage pattern. Moreover, once the shift from aerobic to anaerobic conditions is achieved, fermentation is responsible for the significant ethanol, acetaldehyde, and lactic acid accumulation accountable for off-odours/flavours, tissue damage, and pH drop [56]. These dynamics support the adverse effects observed on the sensory quality and acidification of the FC carrots during storage (Rejection and pH vectors, Figure 2a). The formation of slime, texture changes, and off-flavours are reported as the most common impairments leading to sensory rejection of the product [56]. The high respiration levels and the prevailing lactic flora in FC carrots further boost this deterioration pattern. Thus, the found relation between O_2_, CO_2_, LAB, sensorial rejection, and pH can be used as a proxy for FC carrot quality deterioration. 

PC2 describes the phenolic synthesis dynamic (represented by TPC, CA, and PAL variables (Table 2) and accounting for 29.6% of data variability). The positive correlation represents the known mechanism of phenolic biosynthesis induced by stress factors (heat, UV, and wounding) involving the activation of PAL, a key regulatory enzyme of the phenylpropanoid mechanism, affecting phenolic accumulation, mainly CA in FC carrots [25,40]. Moreover, the weak TPC, CA, and PAL vector projections on PC1 indicate that changes in these variables should not have any sensorial implications.

Variable loadings on PC1 and PC2 measure independent mechanisms responsible for FC carrot quality changes during storage: PC1 refers to sensorial quality loss promoted by microbial growth, and PC2 describes the phenolic synthesis dynamic. Interestingly, both mechanisms are influenced by O_2_ and CO_2_ concentrations, i.e., MAP conditions established by packaging film. The phenolic synthesis mechanism’s oxygen-dependence is suggested by projections of the TPC, CA, PAL, and O_2_ vectors on the first quadrant (Figure 2a). Thus, a consistent trend in the loading plot is shown (Figure 2a): within the interdependence of O_2_/CO_2_ concentrations, microbiological growth is directly influenced by CO_2_ concentration, while changes in O_2_ concentration are influential to the phenolic synthesis dynamic [57]. The modified atmosphere inside packaging suitable for preserving the quality of FC products is widely reported [7,58,59,60,61]. Optimal internal atmosphere conditions are product-specific and depend on several factors, such as respiratory activity, microbial growth, and permeability of the packaging films [7,58,59,60,61]. Particularly for FC carrots (shredded), the preservation of product quality (and its shelf-life) is critical considering the high respiratory levels, the accelerated microbial growth, and the limited permeability of the packaging films to oxygen [7,59]. The interdependence of the variables outlined in the PCA highlights the importance of combining treatments that simultaneously interfere with these mechanisms (multi-target approach).

The projection of analysed samples in the score plot (Figure 2b), from right to left, reflects the increasing quality loss and sensorial rejection of the samples that is accentuated with storage, regardless of the type or combination of treatments applied.

The score plot indicates sample grouping as revealed by the hierarchical cluster analysis dendrogram (Appendix A Appendix A). According to sample grouping, the first division into groups A and B is related to the first mechanism identified over PC1. This dynamic reflects the pronounced microbiological growth and the significant changes in the internal atmosphere of the package (towards anaerobic conditions) as the leading cause of FC carrot quality deterioration. At a second level (groups A1, A2, B1, and B2), the treatments and packaging film effects over the two independent mechanisms responsible for FC carrot quality changes are highlighted. Such a sample clustering indicates that integrated heat-shock technological solutions (single or in combination with UV-C) under the headspace conditions provided by package B (micro-perforated) promote bioactivity enhancement and quality retention of FC carrots. Enhancement of the film permeability to enable O_2_ inlet (as in the case of film B by micro perforation) proved essential for preventing microbial deterioration and phenolic accumulation.

Further exploring the sample distribution (Figure 2b), the UV-treated (U) and untreated (C) samples lie very close to each other when considering sampling dates (from days 0 to 14). A lack of effect from the UV treatment in effectively reducing the initial microbial load and in controlling microbiological growth is suggested.

The effects of heat shock in reducing the initial microbial load (day 0) are revealed by the positioning of the heat-treated samples (H and HU) to the right of the untreated (C) and UV-treated (U) samples. On the other hand, the relative distribution of the heat-treated samples (H and HU) in PC1 shows the significant effects of heat shock in controlling microbial development until day 14. This sample placement also reveals an effect on the delay in changing the composition of the internal atmosphere (O_2_/CO_2_), indirectly expressing a decrease in the product’s respiration rate. Several studies have demonstrated the positive effects of heat shock on the decrease in respiration rate and the decontamination of multiple FCs, preventing MAP conditions favourable to anoxia metabolism [15,16,22,24,62].

Considering the axis defined by PC2, a distinction between samples packaged with films A and B is observed (negative and positive sections of the axis, respectively), demonstrating that a threshold oxygen concentration is necessary for phenolic synthesis (irrespective of the stress applied). The gas composition inside the packages has also been reported to have negative consequences on phenolic synthesis since, when excessively low O_2_ levels are reached, a decrease in phenolic accumulation is registered in fresh-cut products, including carrots [53,55,63,64]. During storage, anoxic conditions (O_2_ levels < 0.5%) were reached in the early stages of storage in samples packaged with film A (Appendix A Appendix A), which could have significantly influenced the phenolic levels achieved (maximum TPC levels ranging from 81.2 to 123.7 mg CAE/100 g). This O_2_ condition never occurred in samples packaged with film B (Appendix A Appendix A), which provides adequate O_2_ levels for sustained and significant phenolic synthesis/accumulation during storage (maximum TPC levels ranging from 111.9 to 153.1 mg CAE/100 g). Thus, oxygen is required for promoting stress-induced phenolic synthesis and accumulation, as also concluded by [65] in FC pineapple and [66] in cherimoya fruits.

Regarding the quality parameters not included in the PCA, during storage, relative maintenance of TCC levels was observed irrespective of sample type as only promptly significant differences were found (Figure 3). Contrary to the expectations and results in other studies (e.g., immature green tomato—1–8 J/cm^2^ [32] and bell pepper—6 kJ/m^2^ [29]), the increase in carotenoid content by UV stress response was not observed. The lack of effect might be attributed to considerable differences in the UV doses applied and in the botanical structures.

The abiotic stress treatment and packaging film effects on the AOx of fresh-cut carrots during storage are shown in Figure 4. The gaseous composition inside the packages was the most influential factor in AOx changes during storage. In samples packaged with film A, significant increases in the AOx levels were determined from days 5 to 7; however, they were dependent on the treatment applied. The heat-treated (H) samples registered the highest AOx increase, followed by the combined-treated samples (HU) and the UV-treated (U) samples, with similar levels to that of the raw material. However, it should be noted that the significant increase in AOx registered in the H samples compared with the C samples is not justified by the increase in TPC (Appendix A Appendix A), both with similar TPC contents (*p* > 0.05). Likewise, the non-significant variations in TCC do not justify this increase either. Conversely, the AOx increases evaluated in the HU samples agree with the evaluated TPC changes (Appendix A Appendix A). On day 10, a decrease (*p* < 0.05) was registered in the HU samples, with similar AOx levels to all samples, just below the raw material, with no further changes during the remaining storage period.

Similar behaviour was found in samples packaged with film B. Notwithstanding, the AOx increment to day 7 was significantly higher than the one registered in samples packaged with film A. No differences were found between the heat-treated (H) and combined-treated (HU) samples, which had the highest (*p* > 0.05) AOx increase (61% above the raw material), or between the UV-treated (U) and untreated (C) samples. Moreover, despite the decrease (*p* < 0.05) in AOx on day 10, the sample AOx levels were similar to those of the raw material (Table 1) until the end of storage (day 14), regardless of the treatment applied. Other studies have reported the opportunity to increase AOx (directly correlated with phenolic biosynthesis) in fresh produce induced by different stress types [18,19,20,25,26,35,41]. In carrots subjected to heat stress, a three-fold increase was observed (at the end of 7 days at 5 °C [67]).

The significant differences in AOx between films A and B may be justified by the O_2_ concentrations available. Indeed, as previously mentioned, the lower O_2_ concentrations in the samples packaged with film A suggest a limitation in phenolic biosynthesis, a determinant in AOx. 

Concerning soluble solids content (SSC), no effect was attributed to stress application, single or in combination, as no significant changes were found regarding the raw material (8.3 ± 0.1 °Brix; Table 1). During storage, no significant changes (*p* > 0.05) in SSC were found either regarding the stress treatments applied or the film used for packaging: SSC varied between 8.1 and 8.7 °Brix (day 0), and between 7.3 and 9.1 °Brix (day 14).

The initial mean Y&M counts were 3.2 Log_10_ (raw material, Table 1), and after processing (day 0), only the untreated (C) and UV-treated (U) samples registered Y&M counts within the same order. The heat- (H) and combined-treated (HU) samples had counts below the quantification limit (<10^1^ CFU/g), consistent with previous results [15,28] and demonstrating the heat-sensitivity of this microbial group and recognised destruction at treatment temperatures above 60 °C–70 °C [68]. The limited effect of UV-C treatments on this microbial group has also been reported in FC watermelon (Δ2.4–7.2 kJ/m^2^ [34]), FC pineapple (Δ200–4800 J/m^2^ [69]), and fresh-cut faba beans (3 kJ/m^2^ [70]).

During storage, no differences (*p* > 0.05) were found between Y&M counts of untreated (C) samples packaged with films A or B (Appendix A Appendix A), indicating that the established gaseous composition inside the packages did not affect this microbial group. On the other hand, only the abiotic stress treatment of heat shock significantly affected Y&M growth. Heat-treated samples (H and HU) recorded Y&M counts below the limit of quantification (<10^1^ CFU/g) throughout the storage period. The effectiveness of the 100 °C/45 s heat shock in eliminating this heat-sensitive microbial group confirms previous findings [15,16,28]. In contrast, Y&M counts ≈5 Log_10_ CFU/g were recorded in UV-treated samples after 14 days with no differences from control samples (C).

The hurdle approach confirms that the 100 °C/45 s heat shock treatment has a multi-target effect of reducing the incidence of bacterial proliferation, improving the bioactive quality, and preserving the sensorial characteristics. These effects are further enhanced when combined with a micro-perforated packaging film (film B) to achieve a favourable internal gaseous composition.

## 4. Conclusions

In the framework of fresh-cut carrots, the integration treatment that led to synergistic effects for preserving the product quality was applying heat shock on the carrots before cutting and packaging the shredded product using a micro-perforated film. This technological approach allowed to control microbial growth, to overcome the raw material’s phenolic content, and to maintain carotenoid content during storage. The lower respiration rate of the shredded carrots provided by the heat shock and the higher permeability of the micro-perforated packaging film were critical to establishing a modified atmosphere suitable for maintaining the fresh-like quality of the product for longer. Further research is needed to model the effects of this approach and to compare it with standard minimal processing operations for preserving fresh-cut carrot quality using sodium hypochlorite as a disinfectant.

## Figures and Tables

**Figure 1 foods-11-02422-f001:**
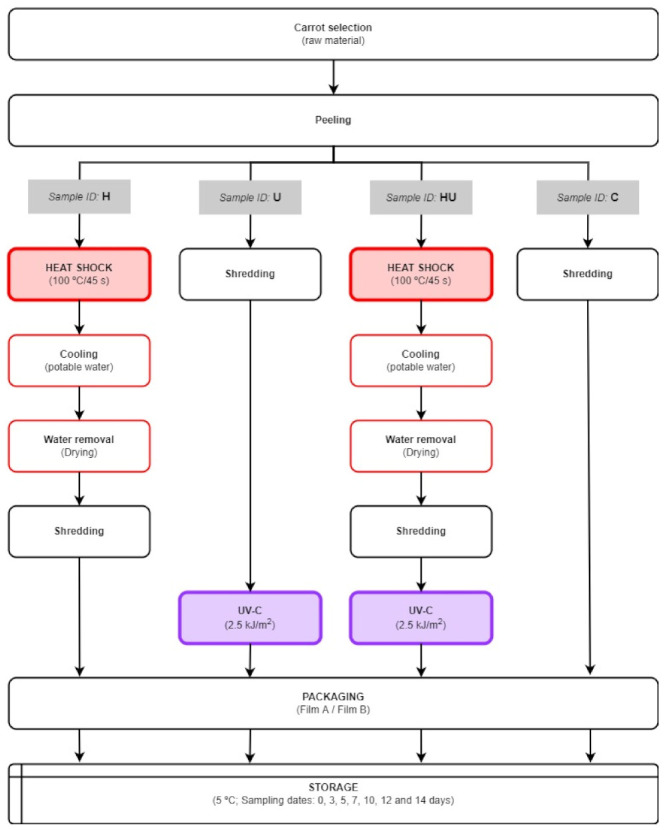
Flow diagram of minimal processing operations for preparing fresh-cut shredded carrots packaged under two MAP conditions according to the selected abiotic stress treatments.

**Figure 2 foods-11-02422-f002:**
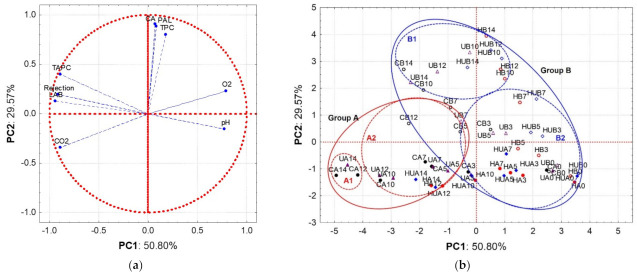
Principal component analysis (PCA) of fresh-cut carrots as affected by abiotic stress treatments and packaging film during low-temperature storage (5 °C, 14 days): (**a**) loading plot and (**b**) score plot.

**Figure 3 foods-11-02422-f003:**
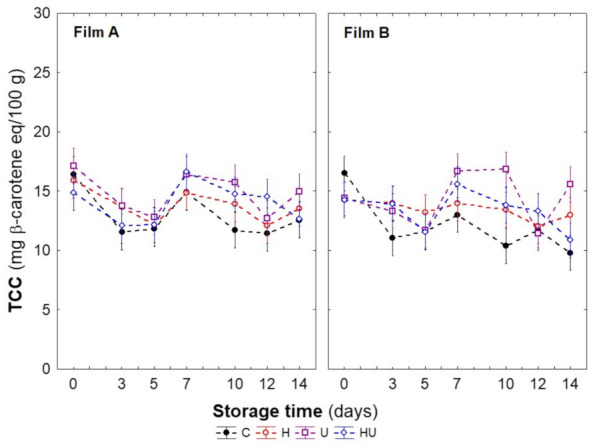
Changes in total carotenoid content (TCC) of fresh-cut carrots as affected by abiotic stress treatments and MAP conditions during low-temperature storage (5 °C, 14 days). Vertical bars denote the confidence interval at 95%.

**Figure 4 foods-11-02422-f004:**
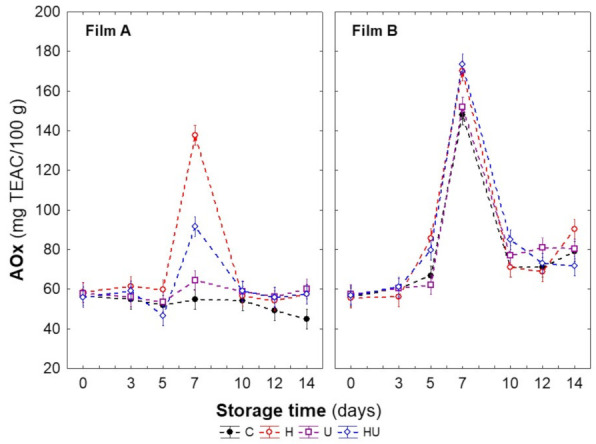
Changes in antioxidant capacity of fresh-cut carrots as affected by abiotic stress treatments and MAP conditions during low-temperature storage (5 °C, 14 days). Vertical bars denote the confidence interval at 95%.

**Table 1 foods-11-02422-t001:** Raw material characterisation (fresh cv. Nantes carrots, unprocessed).

Quality Measurements	Mean ± SD
Bioactive compounds and related responses	
Total phenolic content (TPC, mg CAE/100 g)	85.3 ± 2.2
Chlorogenic acid (CA, mg/100 g)	12.30 ± 1.17
Total carotenoid content (TCC, mg β-carotene eq/100 g)	18.2 ± 0.3
Phenylalanine ammonia lyase (PAL, U/100 g)	28.9 ± 2.5
Antioxidant capacity (AOx, mg TEAC/100 g)	66.7 ± 6.2
pH/SSC	
pH	6.4 ± 0.1
Soluble solids content (SSC, °Brix)	8.3 ± 0.1
Microbial load	
Total aerobic plate count (TAPC, Log_10_[CFU/g])	5.8 ± 0.1
Lactic acid bacteria (LAB, Log_10_[CFU/g])	3.5 ± 0.1
Yeast and mould (Y&M, Log_10_[CFU/g])	3.2 ± 0.2

**Table 2 foods-11-02422-t002:** Factor loadings of the principal component analysis (PCA) based on a correlation matrix between nine quality parameters of FC carrots (56 samples).

Variable	Component 1	Component 2
TPC	0.179766	0.803158
CA	0.071275	0.908997
PAL	0.083203	0.885018
pH	0.774510	−0.154230
Rejection	−0.913445	0.193772
O_2_	0.788420	0.231341
CO_2_	−0.889783	−0.342440
TAPC	−0.889864	0.398211
LAB	−0.942574	0.126346

## Data Availability

The data presented in this study are available on request from the corresponding author.

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
