# Peer review of "Multi-Target Alternative Approaches to Promoting Fresh-Cut Carrots’ Bioactive and Fresh-like Quality"

_foods, 2022, doi:10.3390/foods11162422_

Round 1

Reviewer 1 Report

1. Check the unit of TCC in Figure 3.

2. The TPC and TCC of sample H (in film A) are equivalent to that of control on day 7 (Supplementary Material Table S3 and Figure 3), why is the antioxidant activity of sample H (in film A) about 2-fold higher than that of control on day 7 (Figure 4)? Which compound supports the antioxidant activity in the samples?

3. The antioxidant activity and TPC of control sample are significantly higher in film B than in film A on day 7. The mechanism should be discussed.

Author Response

Manuscript ID: foods-1830129

Type of manuscript: Research paper

Title: Multi-target alternative approaches to promote fresh-cut carrot bioactive and fresh-like quality

Authors: Carla Alegria, Elsa M. Gonçalves*, Margarida Moldão-Martins and Marta Abreu*

Dear Ms. Mia Meng,

We would like to submit the revised version of the following manuscript for your review and consideration in Foods. The manuscript is titled “Multi-target alternative approaches to promote fresh-cut carrot bioactive and fresh-like quality ” authored by Carla Alegria, Elsa M. Gonçalves, Margarida Moldão-Martins and Marta Abreu.

We would like to thank you and the Referees for the review and positive input to our submitted manuscript. We believe in addressing the reviewers’ issues, accepting the suggestions and making alterations as recommended (using the “Track Changes” function in the manuscript). Please find below the responses (in blue) to the Reviewers’ comments.

Thanking you in advance for the consideration given to our work,

Best Regards,

Elsa M. Gonçalves and Marta Abreu

(Corresponding authors)

Responses to the Reviewers’ comments

The responses are given below each reviewer comment.

Reviewer #1

  1. Check the unit of TCC in Figure 3.

We thank the reviewer for pointing out the issue and proceeded accordingly (Line 397).

  1. The TPC and TCC of sample H (in film A) are equivalent to that of control on day 7 (Supplementary Material Table S3 and Figure 3), why is the antioxidant activity of sample H (in film A) about 2-fold higher than that of control on day 7 (Figure 4)? Which compound supports the antioxidant activity in the samples?

Stress-induced phenolic biosynthesis is the primary mechanism acting on AOx of carrot tissues, induced by wounding stress (shredding; C samples) or combined stresses (H samples, wounding stress plus heat shock). In our opinion, the significant changes in antioxidant capacity were, in this study, directly dependent on changes in total phenolic content, as demonstrated by the correlations between AOx and TPC (0.6, p<0.05) and between AOx and TCC (0.2, p>0.05). Indeed, the TCC of the samples remained without significant variations during storage, regardless of the treatments applied. The significant increase in TPC assessed in both samples (C and H) from 0 to 7 days was similar. The nearly 2-fold increase in AOX for H samples is thus not justified and may have resulted from an unexpected error. Therefore, answering the reviewer’s pertinent question, such result should be addressed in the manuscript regarding figure 4 (Lines 408-416). However, we highlight that this particular result does not change any of the paper’s conclusions.

  1. The antioxidant activity and TPC of control sample are significantly higher in film B than in film A on day 7. The mechanism should be discussed.

We have included further discussion to address this issue (Lines 377-387 & 432-435).

Reviewer 2 Report

Fresh-cut fruits and vegetables industry has been developed recently in the world which matches up with the modern busy lifestyles. Carrot has also been marketed as a fresh-cut commodity in different forms, including diced cubes, sticks, grated, and sliced carrots. However, fresh-cuts are usually more perishable which may cause faster loss of sensory and nutritional qualities. Therefore, it is utmost important to practice the pre- and post-harvest technologies to maintain the quality of fresh-cuts. The authors of this research evaluated the heat shock and UV-C radiation as pre-treatments of fresh-cut carrots with the combination of modified atmosphere packaging to maintain the organoleptic and nutritional qualities of fresh-cut carrots.

This study is relevant and interesting to the fresh-cut industry, which will be applicable to maintain their storage quality.

Authors are providing new insight to critical degradation pathways of fresh-cut carrot and promoting bioactive and sensory quality using heat shock and UV-C treatments along with the MAP storage.

Authors discussed the multi-target effects of heat shock treatment on controlling bacterial proliferation, enhancing the bioactive quality, and maintaining the sensory quality attributes of fresh-cut carrots.

The manuscript is well organized and well written.  Authors have used a reader approachable writing style. The conclusion reflects the identified intentions of the study. They have critically and systematically approached their research question.

Dear Authors,

Please, recheck the minor points in your manuscript in regard to subscript/s of CO2 and O2 in the methodology section and results and discussion section.

Author Response

Manuscript ID: foods-1830129

Type of manuscript: Research paper

Title: Multi-target alternative approaches to promote fresh-cut carrot bioactive and fresh-like quality

Authors: Carla Alegria, Elsa M. Gonçalves*, Margarida Moldão-Martins and Marta Abreu*

Dear Ms. Mia Meng,

We would like to submit the revised version of the following manuscript for your review and consideration in Foods. The manuscript is titled “Multi-target alternative approaches to promote fresh-cut carrot bioactive and fresh-like quality ” authored by Carla Alegria, Elsa M. Gonçalves, Margarida Moldão-Martins and Marta Abreu.

We would like to thank you and the Referees for the review and positive input to our submitted manuscript. We believe in addressing the reviewers’ issues, accepting the suggestions and making alterations as recommended (using the “Track Changes” function in the manuscript). Please find below the responses (in blue) to the Reviewers’ comments.

Thanking you in advance for the consideration given to our work,

Best Regards,

Elsa M. Gonçalves and Marta Abreu

(Corresponding authors)

Responses to the Reviewers’ comments

The responses are given below each reviewer comment.

Reviewer #2

Fresh-cut fruits and vegetables industry has been developed recently in the world which matches up with the modern busy lifestyles. Carrot has also been marketed as a fresh-cut commodity in different forms, including diced cubes, sticks, grated, and sliced carrots. However, fresh-cuts are usually more perishable which may cause faster loss of sensory and nutritional qualities. Therefore, it is utmost important to practice the pre- and post-harvest technologies to maintain the quality of fresh-cuts. The authors of this research evaluated the heat shock and UV-C radiation as pre-treatments of fresh-cut carrots with the combination of modified atmosphere packaging to maintain the organoleptic and nutritional qualities of fresh-cut carrots.

This study is relevant and interesting to the fresh-cut industry, which will be applicable to maintain their storage quality.

Authors are providing new insight to critical degradation pathways of fresh-cut carrot and promoting bioactive and sensory quality using heat shock and UV-C treatments along with the MAP storage.

Authors discussed the multi-target effects of heat shock treatment on controlling bacterial proliferation, enhancing the bioactive quality, and maintaining the sensory quality attributes of fresh-cut carrots.

The manuscript is well organized and well written. Authors have used a reader approachable writing style. The conclusion reflects the identified intentions of the study. They have critically and systematically approached their research question.

Dear Authors,

Please, recheck the minor points in your manuscript in regard to subscript/s of CO2 and O2 in the methodology section and results and discussion section.

We thank the reviewer for pointing out the issue and proceeded accordingly.

Reviewer 3 Report

In this work " Multi-target alternative approaches to promote fresh-cut carrot bioactive and fresh-like quality", the authors evaluated the combined effects of eco-friendly treatments (heat shock; UV-C) and MAP conditions on controlling FC shredded carrot key degradation pathways and promoting its bioactive and sensory quality. The authors have worked on an interesting area of research. Specific points are commented on below.

1. Abstract: Add some key quantitative data in the abstract to make it more informative.

2. Line 111: describe how the samples were transported to the laboratory.

3. Line 120: Add CAS number of reagents

4. Line 160: Were UV treatment conditions based on any studies? Cite the source or explain how these conditions were established.

5. 179: further detail on the Headspace gas analysis methodology

6. Results and discussions: This part needs significant improvement. Some of the results are just reported without further discussion.

7. Figure 3: The results could be better visualized on a bar graph with the averaging test.

Author Response

Manuscript ID: foods-1830129

Type of manuscript: Research paper

Title: Multi-target alternative approaches to promote fresh-cut carrot bioactive and fresh-like quality

Authors: Carla Alegria, Elsa M. Gonçalves*, Margarida Moldão-Martins and Marta Abreu*

Dear Ms. Mia Meng,

We would like to submit the revised version of the following manuscript for your review and consideration in Foods. The manuscript is titled “Multi-target alternative approaches to promote fresh-cut carrot bioactive and fresh-like quality ” authored by Carla Alegria, Elsa M. Gonçalves, Margarida Moldão-Martins and Marta Abreu.

We would like to thank you and the Referees for the review and positive input to our submitted manuscript. We believe in addressing the reviewers’ issues, accepting the suggestions and making alterations as recommended (using the “Track Changes” function in the manuscript). Please find below the responses (in blue) to the Reviewers’ comments.

Thanking you in advance for the consideration given to our work,

Best Regards,

Elsa M. Gonçalves and Marta Abreu

(Corresponding authors)

Responses to the Reviewers’ comments

The responses are given below each reviewer comment.

Reviewer #3

In this work "Multi-target alternative approaches to promote fresh-cut carrot bioactive and fresh-like quality", the authors evaluated the combined effects of eco-friendly treatments (heat shock; UV-C) and MAP conditions on controlling FC shredded carrot key degradation pathways and promoting its bioactive and sensory quality. The authors have worked on an interesting area of research. Specific points are commented on below.

  1. Abstract: Add some key quantitative data in the abstract to make it more informative.

We thank the suggestion. Key quantitative data (decontamination efficiency expressed as the reduction in initial microbial counts and total phenolic and chlorogenic acid contents expressed as a percentage of increase) were added to the abstract (Lines 30-33).

  1. Line 111: describe how the samples were transported to the laboratory.

The initially submitted manuscript indicated that carrots were transported to the laboratory in a refrigerated truck. Following the reviewers’ suggestion and to be more informative, we added the temperature range of the refrigerated truck (5 °C±2 °C) and used containers for raw material transportation (Lines 112-114).

  1. Line 120: Add CAS number of reagents

The reagents CAS numbers were added (Lines 121-122).

  1. Line 160: Were UV treatment conditions based on any studies? Cite the source or explain how these conditions were established.

Thank you for pointing out this issue to us. The source was added (Line 175 and Lines 614-615).

The results explored in the ms. are part of the first author’s PhD thesis unpublished results where optimisation and validation of the UV-C dose was pursued. Response surface methodology (RSM) was used to optimise the UV-C treatment conditions (DUV-C dose: 0.1 – 5 kJ.m-2; DStorage time: 0 – 8 days), resulting in an optimal dose of 2.5 kJ.m-2 to achieve the maximisation of total phenolic content during storage of shredded carrot. Shredded carrot samples (125 g) were set into 4-L clear glass jars (closed and vented every 8 h to avoid CO2 accumulation) and stored at 5 °C ( as described in Alegria et al. 2016 and Alegria et al., 2021). The 2.5 kJ.m-2 UV-C dose was subsequently validated, and treated FC carrot samples achieved significant phenolic accumulation during storage through the activation of PAL, surpassing the effects of the wounding stress (control).

  1. 179: further detail on the Headspace gas analysis methodology

Further detail on headspace gas analysis was added (L181-187).

  1. Results and discussions: This part needs significant improvement. Some of the results are just reported without further discussion.

We thank the reviewer for the comment, adding further result discussion (e.g., Line 293-306; Line 377-387; line 432-435).

  1. Figure 3: The results could be better visualized on a bar graph with the averaging test.

We have proceeded in accordance with the reviewer's suggestion (changing the graph of figure 3 to a bar graph, and consequently that of figure 4), but we consider that such a change makes it difficult to interpret it, particularly when comparing the variations occurring between samples (see below).

Alternatively, we considered converting the data in figures 3 and 4 into a table. As it can be confirmed below, the resulting table is quite extensive, making it difficult to interpret any comparison of the evolution of sample quantitative responses during storage of the samples.

Thus, we consider that keeping figures 3 and 4 in the initially submitted format is beneficial for data interpretation.

Tentative figure 3 as a bar graph:

Figure 3. Changes in total carotenoid content (TCC) of fresh-cut carrot as affected by abiotic stress treatments and MAP conditions during low-temperature storage (5 °C, 14 days). Vertical bars denote the confidence interval at 95%.

Tentative figure 4 as a bar graph:

Figure 4. Changes in antioxidant capacity of fresh-cut carrot as affected by abiotic stress treatments and MAP conditions during low-temperature storage (5 °C, 14 days). Vertical bars denote the confidence interval at 95%.

Tentative table with Figure 3 and 4 data:

Table 3. Changes in total carotenoid content (TCC) and antioxidant capacity (AOx) of fresh-cut carrot as affected by abiotic stress treatments and MAP conditions during low-temperature storage (5 °C, 14 days).

Treatment

Packaging film

Storage (days)

TCC

(mg b-carotene Eq.100 g-1)

AOx

(mg TEAC.100 g-1)

C

A

0

16.4fghi±0.3

56.6abcdefgh±4.7

3

11.5abcde±0.4

54.7abcdef±3.4

5

11.8abcde±0.8

51.8abcd±2.2

7

14.9cdefghi±1.8

54.9abcdef±7.8

10

11.7abcde±0.5

54.1abcdef±10.6

12

11.4abcd±0.7

49.1abc±2.8

14

12.5abcdefgh±1.9

44.9a±0.5

B

0

16.5ghi±1.0

56.2abcdef±1.3

3

11.1abc±1.3

60.4bcdefghij±5.9

5

11.6abcde±1.0

66.9efghijklm±0.8

7

13.0abcdefghi±3.4

148.0p±6.1

10

10.4ab±1.0

71.1ghijklmn±2.8

12

11.7abcde±0.4

71.2hijklmn±5.4

14

9.8a±0.5

78.9lmno±0.8

H

A

0

15.9efghi±0.6

58.5abcdefghij±3.7

3

13.6abcdefghi±2.4

61.4bcdefghij±6.6

5

12.1abcdefg±0.7

59.8bcdefghij±1.2

7

14.9cdefghi±2.2

137.7p±6.2

10

13.9abcdefghi±5.1

56.3abcdefg±7.5

12

12.1abcdefg±0.6

54.3abcdef±2.8

14

13.5abcdefghi±1.2

57.5abcdefghi±0.3

B

0

14.3bcdefghi±0.5

55.5abcdef±4.6

3

14.0abcdefghi±0.7

56.2abcdef±3.5

5

13.2abcdefghi±1.0

85.7no±3.3

7

14.0abcdefghi±1.2

170.3q±3.5

10

13.4abcdefghi±1.8

71.1ghijklmn±5.3

12

12.1abcdef±0.6

68.7fghijklm±2.7

14

13.0abcdefghi±0.6

90.4o±0.2

U

A

0

17.1i±0.2

58.0abcdefghi±1.8

3

13.8abcdefghi±2.0

56.1abcdef±2.2

5

12.8abcdefghi±0.6

53.5abcde±2.9

7

16.4fghi±1.5

64.5defghijkl±1.3

10

15.8defghi±0.9

58.7abcdefghij±3.8

12

12.7abcdefghi±0.8

56.1abcdef±2.8

14

15cdefghi±0.4

60.1bcdefghij±0.7

B

0

14.4bcdefghi±0.1

57.5abcdefghi±2

3

13.3abcdefghi±0.3

60.3bcdefghij±3.1

5

11.7abcde±0.8

62.3cdefghijk±4.5

7

16.7hi±1.5

151.9p±2.8

10

16.8hi±0.6

76.9klmno±3.2

12

11.5abcd±1.0

81.1mno±2.0

14

15.6defghi±0.3

80.3mno±1.3

HU

A

0

14.9cdefghi±0.2

55.7abcdef±1.8

3

12.1abcdefg±1.7

59.1abcdefghij±1.9

5

12.2abcdefg±1.3

46.7ab±0.4

7

16.6hi±1.4

91.5o±14.8

10

14.7bcdefghi±0.3

59.1abcdefghij±6.5

12

14.5bcdefghi±0.5

55.7abcdef±1.1

14

12.7abcdefgh±0.9

57.4abcdefghi±1.7

B

0

14.3bcdefghi±0.1

56.9abcdefghi±2.5

3

13.9abcdefghi±1.1

61.2bcdefghij±5.4

5

11.5abcde±0.6

79.7mno±0.6

7

15.6defghi±0.1

173.7q±7.2

10

13.8abcdefghi±0.5

85.1no±3.1

12

13.3abcdefghi±0.3

73.1jklmn±0.7

14

10.9abc±0.6

71.6ijklmn±2.7
